# Cutting the Gordian Knot: Rare Presentation of Nodular Fasciitis as Supraclavicular Swelling with Muscular Involvement in ^68^Ga-FAPI-PET/CT

**DOI:** 10.3390/diagnostics14192238

**Published:** 2024-10-08

**Authors:** Lukas A. Brust, Madeleine Höh, Maximilian Linxweiler, Alessandro Bozzato, Caroline Burgard, Florian Rosar, Katrin Altmeyer, Carl P. Lessenich, Elke Kohlwes, Lukas Pillong

**Affiliations:** 1Department of Otorhinolaryngology, Head and Neck Surgery, Saarland University Medical Center, 66421 Homburg, Germany; myriam-madeleine.hoeh@uks.eu (M.H.); maximilian.linxweiler@uks.eu (M.L.); alessandro.bozzato@uks.eu (A.B.); lukas.pillong@uks.eu (L.P.); 2Department of Nuclear Medicine, Saarland University, 66421 Homburg, Germany; caroline.burgard@uks.eu (C.B.); florian.rosar@uks.eu (F.R.); 3Department of Radiology, Saarland University, 66421 Homburg, Germany; katrin.altmeyer@uks.eu (K.A.); paul.lessenich@uks.eu (C.P.L.); 4Department of Pathology, Mainz University Hospital, 55131 Mainz, Germany; elke.kohlwes@unimedizin-mainz.de

**Keywords:** benign neoplasm, nodular fasciitis, ^68^Ga-FAPI-PET/CT, sarcoma

## Abstract

**Background:** Nodular fasciitis is a benign, singularly occurring nodular fibroblastic/myofibroblastic neoplasia. Due to the rapid growth and cellular atypia, this rare differential diagnosis in the head and neck region can be mistaken for malignant sarcomas. **Methods:** We present a 40-year-old female patient with an unclear, rough, and poorly displaceable supraclavicular swelling on the right as part of a medical check-up. Sonographically, the lump was poorly circumscribed with little vascularization. A consecutive core needle biopsy of the lesion yielded inconclusive results showing spindle-shaped tumor cells. ^68^Ga-FAPI-PET/CT showed an intensive uptake of the right supraclavicular lesion in addition to postoperative changes in the right tonsil. Subsequent operative partial excision of the lesion confirmed the histopathological diagnosis of nodular fasciitis. **Results:** Nodular fasciitis is the most prevalent pseudosarcoma found in soft tissues. This case is the first description of ^68^Ga-FAPI-PET/CT in nodular fasciitis. Surgical removal is advised; nevertheless, the tumor frequently diminishes on its own, and recurrence is rare. Extensive surgical therapy is not necessary. **Conlcusions:** The recognition of nodular fasciitis and its benign characteristics is crucial to prevent diagnostic errors and the subsequent unnecessary operative treatment of the patient.

**Figure 1 diagnostics-14-02238-f001:**
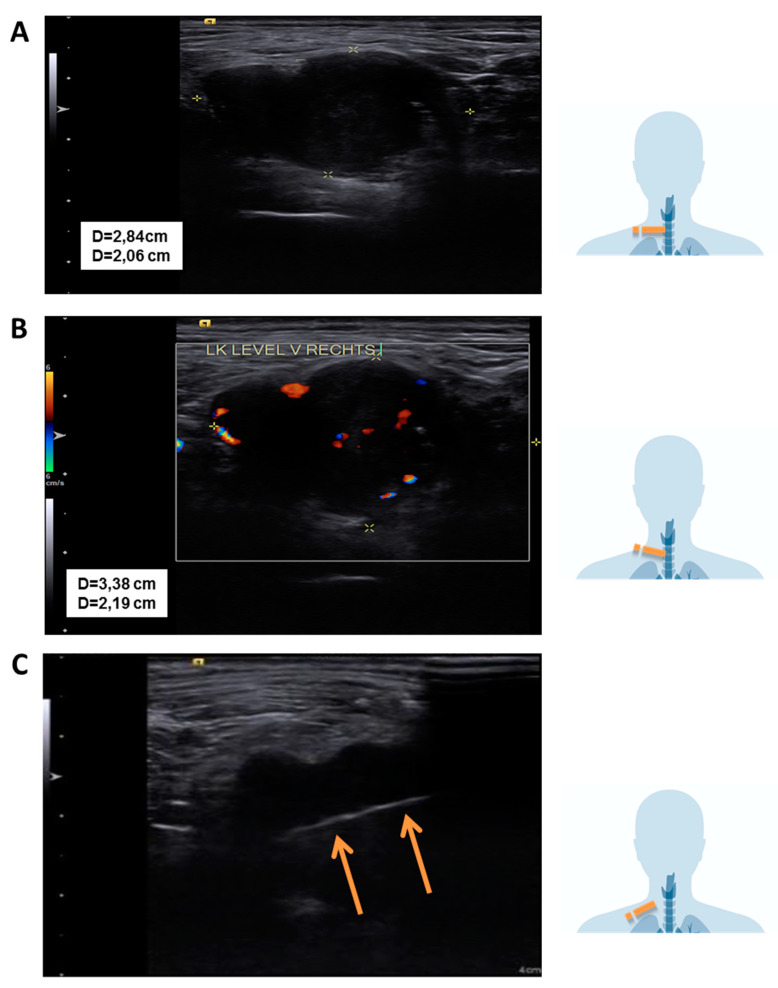
Sonographic findings: (**A**) Sonographic visualization of the lesion using a linear transducer probe at 14 MHz; (**B**) Doppler sonography shows diffuse perfusion. (**C**) In-plane view of the core needle biopsy showing the biopsy channel (orange arrows indicate the intratumoral biopsy tunnel). Pictograms on the right of the image show the sonographic cross section. Nodular fasciitis is a benign, reactive lesion characterized by rapid growth and fibroblastic/myofibroblastic proliferation. It is frequently misdiagnosed as a soft tissue sarcoma due to its rapid, infiltrative growth pattern, high cellularity, and increased mitotic activity [1]. It was first described in 1955 by Konwaler et al. [2] We present a case of a female patient to emphasize the importance of considering this differential diagnosis in the context of rapidly progressing masses in the head and neck area. In our case, a 40-year-old, otherwise healthy female patient presented to our outpatient clinic with an unclear nodular lesion in the right supraclavicular region. The ENT examination findings of the ears, nose, nasopharynx, oropharynx, hypopharynx, and larynx were unremarkable. Sonographically, a coarse, poorly demarcated, peripherally vascularized lesion measuring 3.5 cm × 3 cm was observed. The other cervical lymph nodes and soft tissues of the neck appeared normal in size and configuration. Due to the suspicious finding, a histological verification via core needle biopsy was indicated. This initially revealed an unclear finding with fibrotic skeletal muscle and a broad-based spindle cell neoplasm consisting of disorganized, herringbone-patterned, multipolar, elongated cells with oval, enlarged nuclei and loosened karyoplasm. The initial working diagnosis was a malignant tumor originating from connective tissue.

**Figure 2 diagnostics-14-02238-f002:**
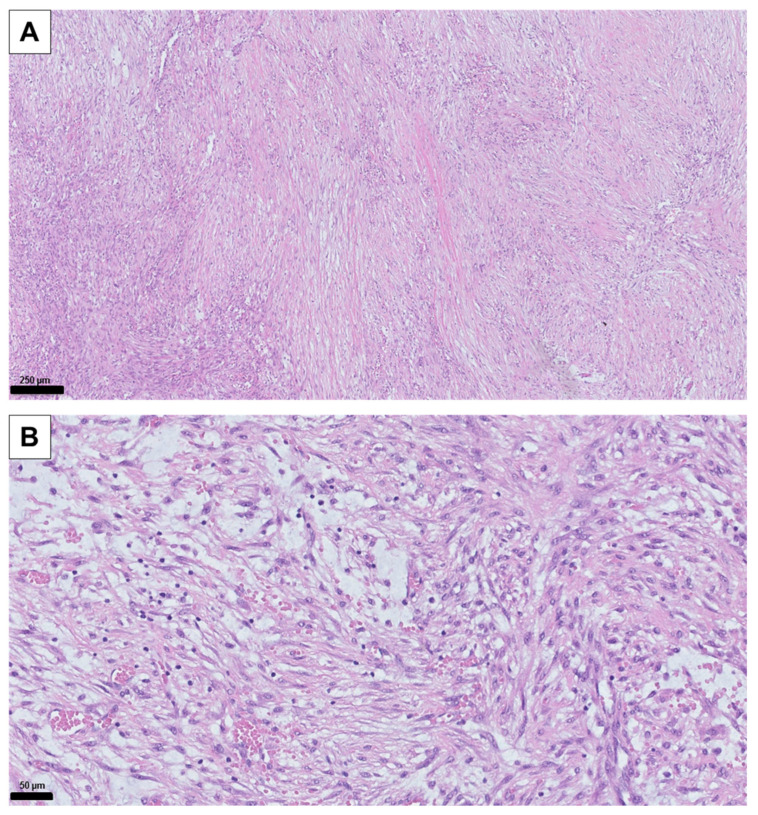
Histological HE staining of the tissue samples: (**A**) 10× magnification of the surgical tissue sample. (**B**) 40× magnification of the surgical tissue sample. Typical variable myxoid changes and extravasated erythrocytes as well as elongated cells with oval, slightly enlarged cell nuclei and loosened karyoplasm are visible. The core needle biopsy was technically challenging due to the anatomical location near the brachial plexus and lung apex. Initial light microscopy findings were nonspecific, raising the possibility of a sampling error where only non-representative peripheral areas of the lesion may have been obtained. Therefore, definitive histological confirmation through open biopsy was pursued, as shown in Figure 2.

**Figure 3 diagnostics-14-02238-f003:**
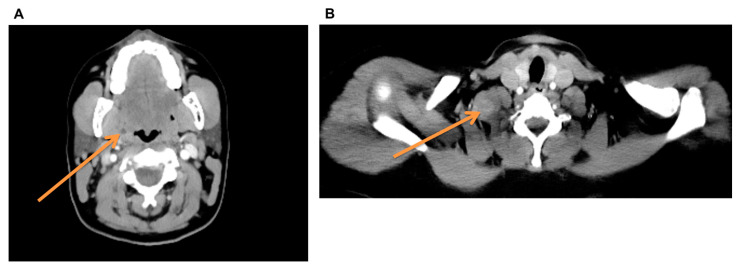
Preoperative CT staging: (**A**) The arrow points to an intratonsillar lesion; the differential diagnosis initially included an abscess and a malignant, centrally necrotizing lesion. (**B**) The arrow points to the radiologically suspicious supraclavicular mass with muscular infiltration in the sternocleidomastoid muscle. The subsequent CT staging examination revealed, in addition to the already sonographically depicted 3.5 cm × 3 cm supraclavicular mass, a hypodense 5 mm intratonsillar lesion on the right side. This constellation of findings supported the suspicion of a possible p16-positive tonsillar carcinoma with supraclavicular lymph node metastasis, although this would have been an atypical metastasis localization. Based on these findings, diagnostic panendoscopy with biopsy of the right tonsil and sonographically guided excision of the right supraclavicular mass was indicated during the interdisciplinary head and neck tumor conference. The intraoperatively obtained frozen section from the tonsil could not exclude malignancy (Figure 4). Subsequently, exploratory excision of the supraclavicular mass was attempted. Intraoperative exploration of the surgical site revealed the full extent of the tumor and its infiltrative growth pattern of the lesion in the sternocleidomastoid muscle, brachial plexus, and apical pleura. Complete removal was impossible due to the infiltrative growth into the brachial plexus and pleura apex. The lesion was reduced by half, preserving all adjacent structures, and a new representative histological sample was taken.

**Figure 4 diagnostics-14-02238-f004:**
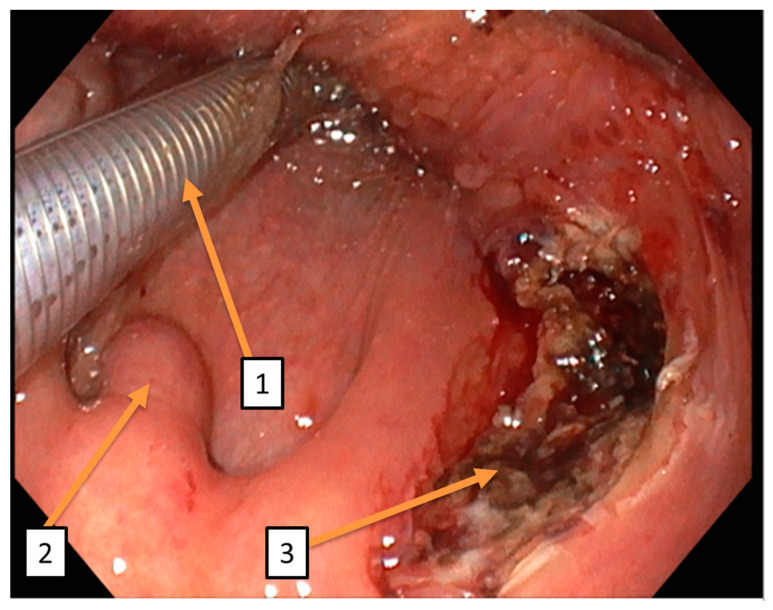
Intraoperative enoral findings: (1) shows the enoral course of the tube. (2) shows the median uvula. (3) shows the intraoperative findings of the right tonsil lobe after trial excision. Given the unresolved nature of the lesion’s dignity, the indication for ^68^Ga-FAPI-PET/CT was determined during the interdisciplinary head and neck tumor conference decision. This imaging modality was employed both for further lesion characterization and as part of pre-therapeutic staging. Comprehensive tumor assessment was conducted to identify potential metastases not detected by conventional CT staging. This showed intensive uptake in the previously described supraclavicular lesion and a reactive tracer accumulation at the biopsy site of the right tonsil. No other systemic lesions were detected (see Figure 5). The definitive histological RNA-based NGS analysis provided evidence of the MYH9-USP6 fusion. Recent studies suggest that the fusion of the MYH9-USP6 genes leads to the overexpression of the USP6 protein in 83–92% of cases, which acts as a pro-oncogene [3,4]. Microscopic histomorphological imaging confirmed the diagnosis of nodular fasciitis. The final findings of the oral lesion ruled out malignancy without direct evidence of nodular fasciitis. Further radical surgical or systemic therapies were avoided. Spontaneous regression after incomplete excision can be expected, as described by Nishi et al. [5]. The patient was thoroughly informed about the benign nature of the disease and the good spontaneous course. The patient was doing well and had not developed any sensory or motor deficits at any time.

**Figure 5 diagnostics-14-02238-f005:**
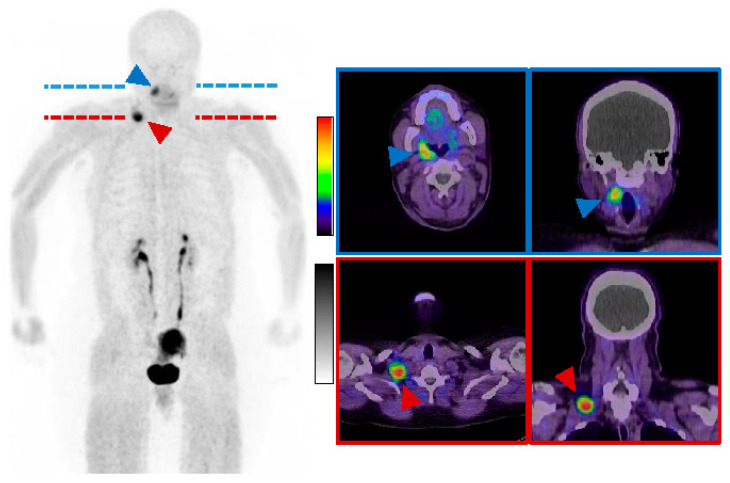
Postoperative ^68^Ga-FAPI-PET/CT: 14 days after the operation, a whole-body PET/CT was performed with 167 MBq ^68^Ga-FAPI-04. Images were acquired 16 min post-injection (2.5 min per bed position). The examination showed a tracer accumulation intratonsillar on the right (blue arrowheads), primarily in the context of postoperative changes. Intense uptake in the supraclavicular mass on the right (red arrowheads); otherwise physiological tracer distribution (among others in the uterus); (**left**): maximum intensity projection (MIP); (**right**): representative transversal and coronal slices. Our case presents the course of a patient with a rapidly progressive mass in the head and neck region. Due to initially unclear findings and the more common malignant lesions in this area, malignant differential diagnoses such as sarcoma and squamous cell carcinoma were considered. The subsequent diagnostic and surgical procedures represented a major psychological burden for the patient and her family. Other working groups reported comparable diagnostic workups [6]. If there is doubt about excluding malignancy, complete removal of the tumor along with selective neck dissection can be a safe option with minimal complications. In addition to the more common subcutaneous localization, intraoral localizations of nodular fasciitis are much less frequently described in the literature [7,8]. The patient’s intratonsillar lesion could represent a second manifestation, although a clear histological confirmation could not be obtained. In summary, the radiological and nuclear medicine findings, along with histological examination, confirmed the diagnosis. The poor sonographic demarcation and muscular infiltration of the lesion are consistent with clinical and radiological findings described in the literature [9]. For FDG-PET/CT, it has been shown that nodular fasciitis is difficult to differentiate from malignant diseases due to its increased glucose uptake and rapid proliferation rate [10]. The intense FAP expression in our case provides additional insights into the use of this imaging modality concerning nodular fasciitis. While FAP is expressed in multiple malignancies, it is generally not detectable in normal tissues [11]. However, the reactive expression of FAP has been described in the context of wound healing [11] processes and diseases with increased fibrosis [12] and can be detected in many benign findings in ^68^Ga-FAPI-PET/CT [13]. The significant uptake in ^68^Ga-FAPI-PET/CT in our case may indicate high fibroblast activity and support the histological findings. The intense FAP expression of the right supraclavicular mass could also be explained by postoperative changes due to the relatively short period of time between surgery and PET scan. Nevertheless, the homogeneous FAP expression suggests that the tumor cells are FAP-positive. To our knowledge, this is currently the only publication in which nodular fasciitis is described as a FAP-positive tumor lesion in ^68^Ga-FAPI-PET/CT. The observations in ^68^Ga-FAPI-PET/CT also point to a diagnostic challenge in nodular fasciitis, as presumed benign and malignant lesions cannot always be differentiated with this imaging modality. This interesting image should draw attention to the fact that nodular fasciitis is a potential pitfall in the staging of malignant disease using ^68^Ga-FAPI-PET/CT.

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
