# Peer review of "Cutting the Gordian Knot: Rare Presentation of Nodular Fasciitis as Supraclavicular Swelling with Muscular Involvement in 68Ga-FAPI-PET/CT"

_diagnostics, 2024, doi:10.3390/diagnostics14192238_

Round 1

Reviewer 1 Report

Comments and Suggestions for Authors

I would recommend the authors to give more details about the performed FAPI PET, in particular, which FAPI used exactly, at what time after surgery, which dose, what time between injection and acquisition, length of acquisition, etc...

Despite the radiotracer uptake I'm not convinced of its specificity towards fibroblasts, in particular being a post-surgery PET scan. As FAP can be highly expressed by tumor cells themselves, and not only by macrophages, how do the authors conclude on the interest of 68Ga-FAPI PET for differential diagnosis between benign and malignant lesions, complementary to histology ?

Author Response

Reviewer 1:

I would recommend the authors to give more details about the performed FAPI PET, in particular, which FAPI used exactly, at what time after surgery, which dose, what time between injection and acquisition, length of acquisition, etc...

Response: We thank the reviewer for this helpful comment. We have added the information in the captions of Figure 5: [“14 days after the operation, a whole-body PET/CT was performed with 167 MBq 68Ga-FAPI-04. Images were acquired 16 minutes post injection (2.5 minutes per bed position).”, Page 6.]

  1. Despite the radiotracer uptake I'm not convinced of its specificity towards fibroblasts, in particular being a post-surgery PET scan. As FAP can be highly expressed by tumor cells themselves, and not only by macrophages, how do the authors conclude on the interest of 68Ga-FAPI PET for differential diagnosis between benign and malignant lesions, complementary to histology?

Response: We thank the reviewer for this helpful comment. We agree that FAP expression of the lesion may also be differential diagnostic due to postoperative changes, as FAPI-PET/CT would be performed 14 days after surgery. However, since the lesion expresses FAP very homogeneously, it rather indicates that the tumor cells are FAP-positive. We also agree that FAPI-PET/CT cannot differentiate between benign and malignant genesis in this case. With this interesting PET/CT image, we also wanted to point out the differential diagnosis of nodular fasciitis in FAP-positive malignancy-suspected lesions. We have clarified this in the manuscript. [“The intense FAP expression of the right supraclavicular mass could also be explained by postoperative changes due to the relatively short period of time between surgery and PET scan. Nevertheless, the homogeneous FAP expression suggests that the tumor cells are FAP-positive.”…” To our knowledge, this is currently the only publication in which nodular fasciitis is described as a FAP-positive tumor lesion in 68Ga-FAPI-PET/CT. The observations in 68Ga-FAPI-PET/CT also point to a diagnostic challenge in nodular fasciitis, as presumed benign and malignant lesions cannot always be differentiated with this imaging modality. This interesting image should draw attention to the fact that nodular fasciitis is a potential pitfall in the staging of malignant disease using 68Ga-FAPI-PET/CT. “ Page 6.]

Reviewer 2 Report

Comments and Suggestions for Authors

Dear Authors,

Thank you for your contribution to the field with this intriguing article. The manuscript discusses the evaluation of nodular fasciitis in the neck region using 68Ga-FAPI-PET/CT. It is noteworthy that the use of 68Ga-FAPI is rapidly becoming more prevalent in clinical settings globally, often outperforming the standard 18F-FDG-PET/CT. This trend is supported by new funding for studies aimed at harnessing the full diagnostic potential of this radiotracer for malignant conditions and their differential diagnoses.

The lesion's high uptake on 68Ga-FAPI-PET/CT aligns well with its histological characteristics, making it an excellent mimicker of other FAPI-positive lesions. However, the sonographic images presented do not align with the described "coarse and poorly demarcated" characteristics; rather, the lesion appears lobulated and heterogeneous hypoechoic, akin to a pathological lymph node with type 3 vascularization.

Regarding the right tonsillar lesion, my primary differential diagnosis would include a retention cyst, as indicated by the absence of peripheral enhancement, rather than an abscess or a centrally necrotic tumor, given the isodensity of the surrounding tonsillar tissue and lack of any expansive effect.

A few points require clarification:

  1. Was a MDT consultation conducted for this patient?
  2. What prompted the use of 18F-FDG-PET/CT post-surgery—was it to further characterize the lesion or as part of a staging process?
  3. Could the limited tissue sample from the core biopsy have precluded the use of next-generation sequencing (NGS)?
  4. Could you clarify what is meant by "exploratory excision"? Is this referring to an open biopsy procedure?

Thank you!

Comments on the Quality of English Language

The manuscript would benefit significantly from thorough revisions for grammar and syntax to enhance clarity and readability.

Author Response

Reviewer 2:

  1. Was a MDT consultation conducted for this patient?

Response: We thank the reviewer for this comment. The case was discussed at our interdisciplinary head and neck tumor conference. Participants in this multidisciplinary team meeting include ENT surgeons, maxillofacial surgeons, radiologists, radiotherapists, pathologists, dermatologists and nuclear medicine specialists. In this context, the CT-graphic images were discussed and evaluated in the clinical context. With an overall inconclusive histology and a previously undescribed intratonsillar lesion, the indication was given for renewed histological confirmation (open biopsy) of the supracalvicular mass and panendoscopy with sampling from the tonsil. We include this information in the appropriate context: [“was indicated during the interdisciplinary head and neck tumor conference”, Page 4.]

  1. What prompted the use of 18F-FDG-PET/CT post-surgery—was it to further characterize the lesion or as part of a staging process?

Response: We thank  the reviewer for this important note. Intraoperatively, it became apparent that a complete resection of the supraclavicular mass was not possible. Due to the still not fully clarified situation of the dignity, the indication for 68Ga-FAPI-PET/CT was made in the context of a new tumor conference decision. On the one hand, it served to further characterize the lesion, on the other hand it served as further staging before initiating a definitive therapy.  A complete diagnosis of the extent of the tumor was obtained in order to visualize possible metastases that could not be detected by CT. A corresponding clarification was added on page five: [“Given the unresolved nature of the lesion's dignity, the indication for 68Ga-FAPI-PET/CT was determined during interdisciplinary head and neck tumor conference decision. This imaging modality was employed both for further lesion characterization and as part of pre-therapeutic staging. Comprehensive tumor assessment was conducted to identify potential metastases not detected by conventional CT staging”, Page 5.]

  1. Could the limited tissue sample from the core biopsy have precluded the use of next-generation sequencing (NGS)?

Response: We thank the reviewer for the relevant comment. The core biopsy cylinders that were generated using the core needle already had adequate material to perform further analyses such as NGS. However, due to the anatomical localization in the immediate proximity of the brachial plexus and the apex of the lung, the core needle biopsy was technically difficult to perform. Due to the initially unspecific light microscopic findings, a sampling error could not be ruled out to the effect that only unrepresentative marginal areas of the lesion were captured by the biopsy. A definitive histological confirmation was pursued. A further specification is attached as follows: [The core needle biopsy was technically challenging due to the anatomical location near the brachial plexus and lung apex. Initial light microscopy findings were nonspecific, raising the possibility of a sampling error, where only non-representative peripheral areas of the lesion may have been obtained. Therefore, definitive histological confirmation through open biopsy was pursued as shown in Figure 2. ,Page 3.]

  1. Could you clarify what is meant by "exploratory excision"? Is this referring to an open biopsy procedure?

Response: We are happy to specify the surgical procedure that is meant by exploratory excision: The CT graphic staging examination was able to achieve a very good anatomical localization of the mass. However, only the exploration of the surgical site revealed the full extent of the tumor and its relationship to the surrounding structures such as brachial plexus branches or the pleural apex. Exploratory excision in this context describes the surgical procedure in which the extent and infiltration are assessed intraoperatively and the tumor is excised in the same setting. A further specification is attached as follows: [Intraoperative exploration of the surgical site revealed the full extent of the tumor and its infiltrative growth pattern of the lesion in the sternocleidomastoid muscle, brachial plexus, and apical pleura. ,Page 4.]

Comments on the Quality of English Language:

The manuscript would benefit significantly from thorough revisions for grammar and syntax to enhance clarity and readability.

Response: We deeply apologize for the typos that have been overseen. By supplementing the relevant passages and explaining the aforementioned aspects, we hope to improve comprehensibility. Additionally, the manuscript has been carefully proofread by a very experienced speaker in order to improve the language quality. 
